# Impact of Thrombocytopenia on Survival in Patients with Hepatocellular Carcinoma: Updated Meta-Analysis and Systematic Review

**DOI:** 10.3390/cancers16071293

**Published:** 2024-03-27

**Authors:** Leszek Kraj, Paulina Chmiel, Maciej Gryziak, Laretta Grabowska-Derlatka, Łukasz Szymański, Ewa Wysokińska

**Affiliations:** 1Department of Oncology, University Clinical Centre, Medical University of Warsaw, 02-091 Warsaw, Poland; 2Department of Molecular Biology, Institute of Genetics and Animal Biotechnology, Polish Academy of Science, 01-447 Magdalenka, Poland; l.szymanski@igbzpan.pl; 3University Clinical Centre, Medical University of Warsaw, 02-091 Warsaw, Poland; 42nd Department of Clinical Radiology, University Clinical Centre, Medical University of Warsaw, 02-091 Warsaw, Poland; 5Division of Hematology and Medical Oncology, Mayo Clinic, Jacksonville, FL 32224, USA

**Keywords:** hepatocellular carcinoma, HCC, thrombocytopenia, platelets, outcome, tumor features

## Abstract

**Simple Summary:**

Currently, hepatocellular carcinoma (HCC) is one of the most prevalent oncological diagnoses worldwide. Despite intensive research into its pathogenesis and clinical course, numerous issues remain unresolved. In the presented meta-analysis and systematic review, we investigate the potential of blood platelet levels in patients with HCC for prognostic assessment. Blood platelets may play a significant role as an easily measurable laboratory parameter in assessing the prognosis of patients with HCC.

**Abstract:**

Background: Platelets (PLT) have a role in the pathogenesis, progression, and prognosis of hepatocellular carcinoma (HCC) and could represent a readily measurable laboratory parameter to enhance the comprehensive evaluation of HCC patients. Methods: The PubMed, Web of Science, and Scopus databases were searched with a focus on survival as well as patient and tumor-specific characteristics in correlation to reported PLT counts. Survival outcomes were analyzed with both common-effect and random-effects models. The hazard ratio (HR) and its 95% confidence interval (CI) from analyzed trials were incorporated. Studies that did not provide survival data but focused on platelet count correlation with HCC characteristics were reviewed. Results: In total, 26 studies, including a total of 9403 patients, met our criteria. The results showed that thrombocytopenia in HCC patients was associated with poor overall survival (common-effect HR = 1.15, 95% CI: 1.06–1.25; random-effect HR = 1.30, 95% CI: 1.05–1.63). Moreover, three studies reveal significant correlations between PLT indices and tumor characteristics such as size, foci number, and etiology of HCC development. Conclusion: Our meta-analysis confirmed that PLT count could act as a prognostic marker in HCC, especially with a PLT count cut off <100 × 10^3^/mm^3^. Further prospective studies focusing on the role of PLT in clearly defined subgroups are necessary.

## 1. Introduction

Primary hepatocellular carcinoma remains one of the most commonly diagnosed malignancies with a male predilection, occupying sixth place in incidence and third in mortality worldwide [1,2]. It is predicted to maintain a steady growth, increasing by 55.0% between 2020 and 2040, with an estimated 1.4 million new diagnoses in 2040 [3]. Multiple risk factors have been established for HCC including chronic hepatitis B (HBV) and C viral (HCV) infections, environmental carcinogen exposure, chronic alcohol consumption, metabolic disorders, and rare genetic alterations. Irrespective of the underlying etiology, liver cirrhosis remains the predominant risk factor for HCC with risk rates of approximately 8.3% at 5 years and 12.2% at 10 years [4]. HCC can be understood as a heterogeneous malignancy with diverse etiologies, risk factors, and clinical course, which makes the management of patients challenging.

The current therapeutic approach in HCC is driven by patient performance status, tumor stage, and liver function, which are incorporated into the Barcelona Clinic Liver Cancer Classification System (BCLC) [5]. The updated version stratifies patients treated with curative intent into those eligible for local-regional therapies as well as those considered to be candidates for liver transplant (LT) [6]. For the purpose of transplantation, qualifying liver function and tumor-specific parameters are referred to as the “extended criteria for liver transplantation” [7]. Patients with advanced or metastatic tumors, as well as those excluded from procedures due to severe liver dysfunction, are eligible for palliative systemic treatment. Currently, this involves a combination of immune checkpoint inhibitors (ICIs) and tyrosine kinase inhibitors (TKIs) [8]. Despite a better understanding of this disease and improvement in diagnostics and treatment, the prognosis for HCC is still poor with an overall 5-year survival rate estimated at approximately 20%, with some studies utilizing mainly TKI reporting rates as low as 9.72% for advanced HCC [2,9].

Cancer research has been focused on establishing effective prognostic and predictive markers to guide therapy over the last several decades [10,11]. Ideally, those markers are minimally invasive, cost-efficient, and highly specific to enable their use in clinical management [12]. Platelets play an important role in both cancer growth and spread [13]; with their significance documented for pathogenesis and prognosis in HCC [14,15]. In a healthy organism, platelets are physiologically part of the inflammatory response, transporting and releasing cytokines such as serotonin, platelet-derived growth factors, and transforming growth factor-β in addition to their role in primary hemostasis [16]. In disease states, platelet quantitative and functional indices show a strong correlation with hepatic fibrosis and cirrhosis, precursor states to HCC development [17,18,19]. Consequently, the platelet count can be used with other inflammatory markers in scoring indexes for HCC, such as the AST to Platelet Ratio Index (APRI) or Platelet-Albumin-Bilirubin Index (PALBI) [20,21,22], with studies showing worse overall survival in HCC patients with an index representing a more advanced stage [15,23,24]. Recent analyses of PLT’s role in HCC demonstrated substantial variations depending on the patient cohort characteristics and the geographical context of the research. In contrast to those of preceding investigations, the results of Scheiner et al. showed that thrombocytopenia and platelet activation parameters distinctly correlated with a more favorable prognosis [25]. In a cohort of 378 patients with thrombocytopenia undergoing palliative treatment, defined as <150 g/L, there was an association between thrombocytopenia and advantageous baseline tumor characteristics, including a diminished diameter of the largest nodule, constrained extrahepatic spread, diminished macrovascular invasion, and lower BCLC stages. Moreover, the composite variable of thrombocytopenia and elevated mean platelet volume (MPV), a platelet activation parameter, independently correlated with prolonged overall survival (HR 0.80, 95% CI—0.65–0.98; *p* = 0.029) [25]. However, a subsequent follow up of this work, undertaken on a distinct Taiwanese population, reported markedly dissimilar findings [26]. Despite the thrombocytopenia group manifesting characteristics suggestive of lower tumor aggressiveness in the cohort of over three thousand patients, no observable difference in terms of OS emerged between the groups. Regrettably, data detailing the treatment methodology for these patients were not disclosed [26].

Motivated by these observed differences and taking into account existing meta-analyses [27], our current study endeavored to enhance the understanding of the role of platelet count in HCC prognosis. To achieve this objective, we conducted a comprehensive systematic review and meta-analysis with a subgroup analysis of patients treated with curative and palliative intent. Our working hypothesis was that thrombocytopenia is associated with a worse patient survival independently of the intended treatment modality. Consequently, we reviewed and summarized additional clinicopathological data pertaining to PLT, presenting a consolidated overview herein.

## 2. Materials and Methods

### 2.1. Search Strategy

The aim of the study was to answer the following questions: (1) Does thrombocytopenia affect the survival of patients with HCC? (2) Is there a correlation between thrombocytopenia and the clinicopathological features of the tumor? (3) How does thrombocytopenia correlate with HCC etiology? In order to answer these questions, we followed the Preferred Reporting Items for Systematic Reviews and Meta-analyses (PRISMA) statement guidelines to design the study, analyze the results, and report our findings. A systematic literature review of PubMed, Web of Science, and Scopus databases was performed by two identifications in November 2023. The core search included the following terms: platelets OR PLT OR thrombocytopenia OR thrombopenia AND HCC AND hepatic cancer AND hepatocellular carcinoma AND liver cancer AND prognosis. After the initial search, 4105 articles were found. Eventually, after removing duplicate papers and excluding articles that didn’t meet the inclusion criteria, 26 articles were included in a qualitative synthesis. As one of the included studies developed three separate datasets [28], the rest of our reporting in this meta-analysis will refer to 28 separate studies. Three articles with clinicopathological findings regarding the HCC and PLT count correlation were included in the systematic review to answer our second query but were not included in the meta-analysis results. The literature search included only human studies with no restrictions regarding the year of publication. The authors chose only articles in English. A detailed search strategy is presented in Figure 1. Additional papers were identified by a manual search of the references from the eligible review articles.

### 2.2. Eligibility Criteria

The inclusion criteria were as follows: (1) HCC confirmed in histopathology or by imaging, (2) only original studies, (3) studies cover the topic of HCC patient’s survival or tumor features in relation to platelet count, (4) reported hazard ratios (HRs) and 95% confidence intervals (CIs) for overall survival (OS) or provided sufficient data to calculate these values, and (5) the subjects were divided into low and high PLT level groups or provided data was sufficient to estimate these groups.

Exclusion criteria were: (1) pathological type was not HCC; (2) not relevant to the prognosis or features of HCC; (3) the risk effects and corresponding 95% CI were not provided and available data were not sufficient to calculate them; (4) studies without original clinical data, such as reviews, systematic reviews, meta-analysis, expert opinions, editorial, or comment; (5) clinical trials exclusively evaluating drugs/medical interventions among patients with HCC.

The search results were reviewed by 2 independent researchers (P.C./L.K.) for potentially eligible studies. Disagreements over the eligibility of an article were resolved by consensus.

### 2.3. Data Extraction and Quality Assessment

Relevant information was extracted including author name, publication year, country, study design, sample size, platelet count and cut-off values, data regarding patient survival such as OS, DFS, and postoperative complications, data regarding tumor features such as tumor diameter, and HCC etiology. Where possible we extracted HR with 95% CI; where it was not stated explicitly, we used the parameters from the study to calculate it ourselves. The quality assessment was conducted using the Newcastle–Ottawa scale (NOS) method.

### 2.4. Data Analysis

The pooled HR value for overall survival was calculated using a fixed- and random-effects model. The heterogeneity among the studies was assessed using the Q value and the I^2^ statistic value. A random-effect model was used for data showing statistically significant heterogeneity if *p* < 0.1 as determined by the Q statistic or I^2^ > 50%; otherwise, we considered that there was no obvious heterogeneity and used a fixed-effect model. In cases of high heterogeneity, we conducted a subgroup analysis and meta-regression to explore potential sources of the effect. Furthermore, an influence analysis was performed to assess whether any individual study significantly affected the result. We analyzed those covariates that may have contributed to the potential heterogeneity, for which the PLT cut-off value, treatment intent, presence of HCV infection, number of recruited patients, and Child–Pugh grade were assessed. Eventually, publication bias was examined with Egger’s tests. Statistical analysis was performed and visualized using R version 4.3.2 with the ‘meta’ package [29].

## 3. Results

### 3.1. Characteristics of the Included Studies

The baseline characteristics of the included studies are summarized in Table 1. We were able to identify a cohort of 9403 patients, predominantly male (>50%), with a follow-up duration ranging from 5 months to 5.8 years. PLT cut-off values varied between 75 and 150 (×10^9^/L), with most studies adopting the threshold based on the definition of thrombocytopenia (<100 × 10^9^/L). The studies exhibited moderate to high quality, as reflected by Newcastle–Ottawa Scale (NOS) scores ranging from 4 to 9, with a mean score of 6.54. The studies included predominantly Asian populations. The assessed interventions included curative and palliative therapies; all were analyzed collectively and within subgroups. The hazard ratios (HR) for overall survival (OS) were available in all included studies, providing a comprehensive evaluation of the outcomes. The characteristics of the patient groups from the studies included in the analysis are summarized in Table 2. 

As mentioned before, three additional studies were selectively incorporated into the systematic review due to their intriguing contextual relevance to the clinical-pathological characteristics of HCC based on the patient’s blood platelet count and activation. These supplementary investigations are elucidated in the paragraph following the meta-analysis.

### 3.2. Pooled HR Values for All of the Studies

By accessing general effects, we estimated that the low PLT count in HCC patients was associated with worse OS (HR = 1.30, 95% CI: 1.05–1.63). The forest plot for this estimation is shown in Figure 2. A significant degree of heterogeneity was noted between studies; however, due to consistencies reported in results, all were included in the analysis.

We observed a significant degree of heterogeneity between studies; however, based on our analysis we believe that all of the abovementioned studies evaluated the same methodological approach and decided to interpret them further (I^2^ = 86.0%, *p* < 0.01).

### 3.3. Adjusted Significance of PLTs in HCC

For more accurate results, a separate analysis including an adjusted HR for OS, using Cox multivariate analysis, was performed. Nineteen studies involving 5885 patients were included in this analysis, yielding results that confirmed an unfavorable prognosis (HR: 1.47, 95% CI 1.15–1.86). The forest plot for this estimation is shown in Figure 3.

### 3.4. Pooled HR Values for Various Treatment Groups

Due to significant prognostic heterogeneity in HCC, the studies included in the overall analysis were stratified based on treatment intent. Curative treatment was defined as resection or RFA, while palliative (non-curative) treatment involved TACE and systemic therapy. However, it is important to note that segmental TACE can currently be used with curative intent [55]. We conducted separate estimations for patients treated with curative intent for those receiving palliative interventions. In the cohort undergoing radical treatment, a decreased platelet level indicated a worse prognosis (HR 1.62, 95% CI 1.25–2.11). A different trend was also observed for the palliative care group, which allowed for a contradictory conclusion, where lower PLT levels correlated with better OS (HR 0.81, 95% CI 0.62–1.05). The forest plots for these estimations are shown in Figure 4 and Figure 5.

### 3.5. Exploration of Heterogeneity

As previously mentioned, the estimated HR for OS exhibited considerable heterogeneity despite the inclusion of numerous studies. To elucidate the origins of this heterogeneity, we conducted subgroup heterogeneity analyses. The analyzed covariates encompassed platelet (PLT) cut-off values (150, 101–149, or ≤100), treatment modality (curative vs. palliative), hepatitis C virus (HCV) presence (<50% vs. >50% of patients), total patient enrollment (≥200 vs. <200), and Child–Pugh grade (studies with >50% patients classified as grade A vs. <50%). The findings of this analysis are detailed in Table 3. Our assessment suggests that the treatment intent and the PLT cut-off value may have influenced the pooled effect size (*p* < 0.05 in the subgroup). However, it is important to acknowledge that there are additional unexplored factors that could have impacted our results, especially in a setting of high heterogeneity of the studies. Notably, the PLT level emerged as a significant prognostic factor for patients undergoing curative therapy, with an HR value of (HR 1.62, 95% CI 1.25–2.11). Conversely, for patients undergoing palliative treatment, an association between PLT levels and survival was noted, although a discernible trend towards better prognosis was observed. Also, subgroups with studies that included less than 50% of patients with Child–Pugh A showed no correlation with platelet levels.

### 3.6. Sensitivity Analysis and Test of Publication Bias

A sensitivity analysis was performed to evaluate the reliability of the aforementioned findings. We examined the application of both random-effects and (common) fixed-effects models for the analysis of the studies and observed no discernible disparities between them (comparisons are shown in the corresponding figures). Furthermore, we performed an influence analysis and found that no single study affected the summary estimation (Figure 6). To assess publication bias, Egger’s test was conducted with a funnel plot, shown in Figure 7 (estimated *p*-value of 0.0925).

### 3.7. Additional Clinicopathological Findings

During the literature search, three studies closely approached our inclusion criteria; however, due to the absence of prognostic analyses, they were not included in the meta-analysis [56,57,58]. Nevertheless, their significance in exploring the clinical-pathological features and disease progression as they relate to patient’s platelet counts was noteworthy, leading to their inclusion in this review section. All three studies aimed to establish correlations between commonly assessed parameters in patients with HCC, such as gamma-glutamyl transferase (GGTP), PLT, alpha-fetoprotein (AFP), or bilirubin levels, and characteristics of the tumors, including the size and number of lesions. A lower PLT count was observed primarily in small tumors (≤3 cm) with concurrent higher bilirubin values. This observation seemed contradictory to the potential process of parenchymal destruction in the setting of tumor growth, suggesting compelling hypotheses about new pathways in HCC carcinogenesis [57]. Small tumors were also observed in cirrhotic livers, where their development is constrained by both liver function and parenchymal collagen remodeling. A discernible association was observed between increasing tumor size and elevated platelet levels, perhaps due to the presence of paraneoplastic phenomena. A subsequent cohort analysis revealed that the mean platelet levels were 142, 158, and 239 × 10^9^/L, respectively, corresponding to the progressive increase in tumor size (*p* = 0.0001). Additionally, patients in the cohort with the largest tumors exhibited a significantly lower incidence of cirrhosis. This led to speculation that platelet-releasing mediators and growth factors might in fact contribute to the more aggressive tumor growth and larger sizes, partly addressing the question of interactions between the tumor microenvironment and HCC characteristics [56]. However, after adjusting these parameters for patient survival, no statistically significant correlations were found [58].

## 4. Discussion

Despite the association of platelet count with HCC outcomes, it is not currently directly incorporated into any of the widely used predictive and prognostic tools, with only the indirect inclusion of platelet count as a predictor of portal hypertension [59,60].

Our analysis underscores role of thrombocytopenia in the prognosis of patients with HCC and may offer impetus for the further refinement of prognostic models. Platelet counts below 100 × 10^9^/L increased the overall risk of death by 30% across all patient groups. In individuals undergoing treatment with curative intent, this risk was amplified by 62%. Additionally, an individual analysis revealed an independent association between thrombocytopenia and OS in patients with HCC. Notably, there were no significant differences between the results obtained using random- and fixed-effects models, and the pooled HR value remained largely unaffected by any single study. These outcomes underscore the robustness of our data. Through subgroup analysis, potential sources of heterogeneity were identified, particularly in the context of PLT level cut-off values, warranting further evaluation alongside other covariates. Our analysis may serve as a rationale for further research on the more comprehensive inclusion of PLT in the overall assessment of HCC patients. Additionally, future analyses may be further enriched by incorporating more tumor-specific characteristics in relation to platelet values, providing an additional patient stratification tool.

Somewhat unexpectedly, our analysis revealed that low platelet counts did not influence outcomes in patients treated in a palliative setting. We expected patients with low PLT to be at risk of receiving less systemic therapy as in clinical trials of the newest immunotherapy, targeted therapy, and their combinations, the cut-off value for platelet counts for inclusion oscillated at around ≥60–75 × 10^9^/L [61,62,63]. As the studies we scrutinized utilized a higher cut-off value for the platelet levels compared to the recommended thresholds for patient inclusion in a clinical trial, it is very difficult to gauge whether the reception of therapy would have influenced those results and if results would have been different with lower PLT thresholds.

There were several limitations to our study. First, some included studies did not report important data (such as Child–Pugh grade and HCV infection rates) essential for a comprehensive analysis. Furthermore, there are major obstacles to obtaining a homogenous group of HCC patients. Clinical trial results may be difficult to apply to real-world situations due to incomplete or unclear reporting of data. However, it can also be a starting point for in-depth analysis for the future. Second, most of the included studies investigated the impact of platelet levels among patients undergoing diverse surgical procedures, with a minority focusing on systemic therapy in the form of chemotherapy, immunotherapy, or targeted therapy. Consequently, the number of patients receiving systemic therapy in our analysis was limited. Third, the quality of some studies included in the analysis was moderate, as reflected by a Newcastle–Ottawa Scale (NOS) score of 4. Fourth, the PLT value cut-off points in most studies were set to 100 × 10^9^/L; only four studies considered a lower cut-off point. Consequently, establishing the precise PLT level indicating a worse prognosis remains elusive at present. However, the primary challenge encountered in our analysis pertains to the pronounced heterogeneity, surpassing that observed in previous studies of a similar nature. While it might be attributed to the inclusion of a novel patient group in recent years and the expansion of clinical indications for surgical treatments, including transplantation and hepatectomy, the exact source of this heterogeneity is yet to be determined.

Despite extensive in vitro and in vivo investigations into the relationship between platelets and HCC, certain questions remain unresolved. The evident influence of both thrombocytopenia and thrombocytosis on the pathogenesis of HCC is primarily attributed to the interaction of platelet mediators with a cirrhotic liver and tumor microenvironment [64,65]. It has been established those metabolites released from platelets, including platelet-derived growth factor (PDGF), actively promote tumorigenesis. Within HCC cells, the level of PDGF receptor alpha (PDGFRα) is elevated in comparison to normal hepatocytes [64,66], resulting in heightened chemosensitivity under normo- or hypoxic conditions when PDGF is suppressed [64]. Additionally, tumors exhibiting an overexpression of PDGFRα are significantly associated with increased micro-vessel density, macroscopic vascular invasion, shorter overall survival, and a higher rate of HCC recurrence [67]. Moreover, serotonin, a crucial mediator released from platelets, may impact disease progression. Circulating serotonin levels were notably higher in cirrhotic patients with HCC than in those without HCC [68]. Furthermore, in a cohort of 40 HCC patients undergoing partial hepatectomy, intra-platelet serotonin levels were predictive of HCC recurrence (HR 0.1, 95% CI—0.01–0.89) [69]. Hence, the PLT count, and consequently, levels of mediators released from its granules may exhibit correlations with both cirrhosis and the characteristics and aggressiveness of tumors. The discussed studies exploring the relationship of platelet count and activity with HCC tumorigenesis regrettably could not be incorporated into our meta-analysis owing to discrepancies in the OS assessment criteria employed by the researchers [25]. Nonetheless, our findings exhibit concordance with the aforementioned work, as well as with the divergent commentary concerning that analysis [26]. Consequently, we combined the conclusions from both studies, which showed different prognostic outcomes depending on the patient cohort and type of therapy employed [25,26].

Individual analyses of the impact of PLT on a patient’s prognosis provide valuable information about the prognosis in a specific group of patients. For instance, an analysis that considers only surgically treated patients. However, we have not observed a direct comparison of palliatively and radically treated patients in a trial. We decided to investigate whether the thresholds for inclusion were appropriate, given the strict guidelines for patient inclusion in different treatments. For instance, based on our findings, it may soon be feasible to incorporate individuals with significantly lower PLT levels, who are currently ineligible for treatment, into both clinical trials and routine care. The primary benefit of our research is its clinical relevance, which could offer physicians more choices for patients who experience ongoing platelet abnormalities and related symptoms. The results of the individual analyses showed some inconsistencies. For instance, some analyses indicated that patients with reduced platelets had a better prognosis, while others suggested a worse prognosis. Our work suggests that the type of treatment may be a possible source of this discrepancy.

## 5. Conclusions

There is a strong association between platelet counts and survival outcomes in HCC patients, although a comprehensive incorporation of platelet assessment in the evaluation of HCC patients requires further refinement. This will require the identification of specific subgroups of patients with this heterogeneous cancer where platelet count and other PLT indices may influence pathogenesis and prognosis. Moreover, further prospective trials, especially on patients treated with systemic therapy, are much needed with clear inclusion criteria and comprehensive reporting.

## Figures and Tables

**Figure 1 cancers-16-01293-f001:**
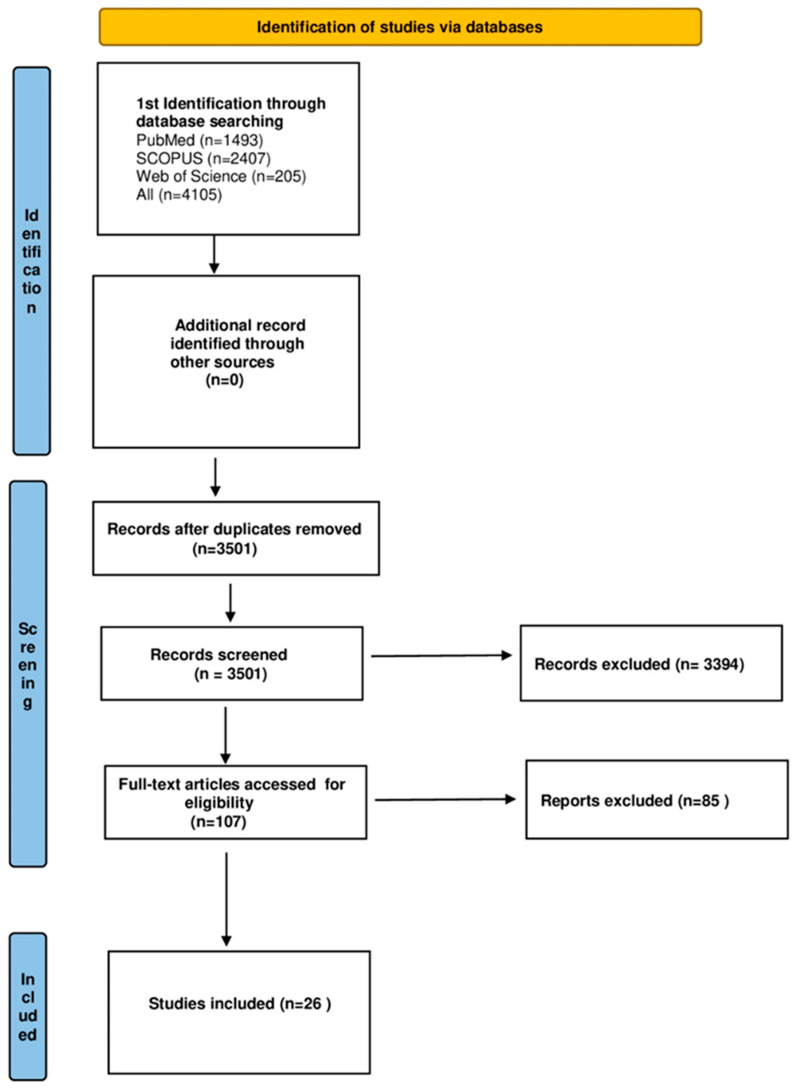
Flowchart presenting the process of article selection, according to Preferred Reporting Items for Systematic Review and Meta-Analyses (PRISMA) guidelines.

**Figure 2 cancers-16-01293-f002:**
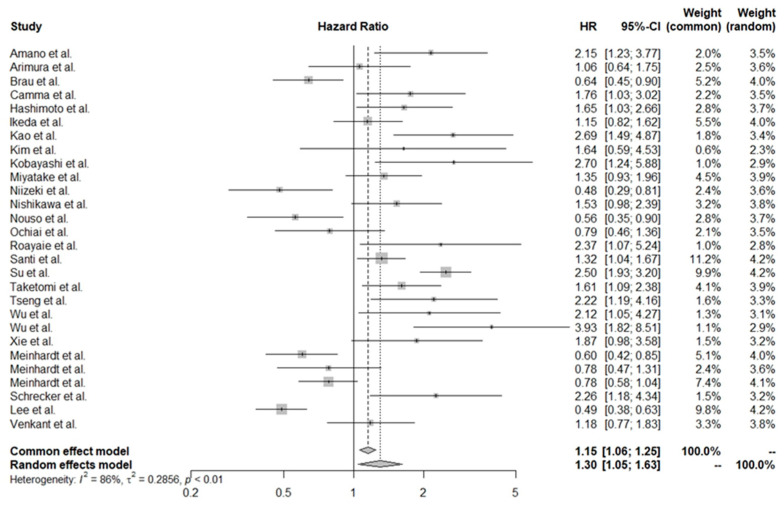
Effect of thrombocytopenia on overall survival in patients with hepatocellular carcinoma [28,30,31,32,33,34,35,36,37,38,39,40,41,42,43,44,45,46,47,48,49,50,51,52,53,54].

**Figure 3 cancers-16-01293-f003:**
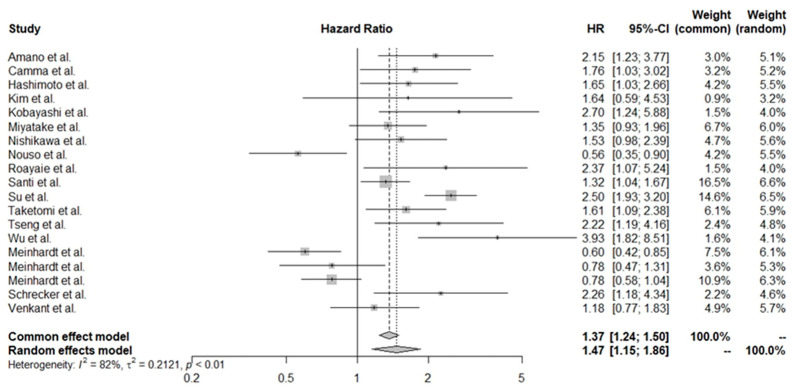
Effect of thrombocytopenia on overall survival in patients with HCC (adjusted) [28,30,33,34,37,38,40,42,43,45,46,47,48,49,50,51,53].

**Figure 4 cancers-16-01293-f004:**
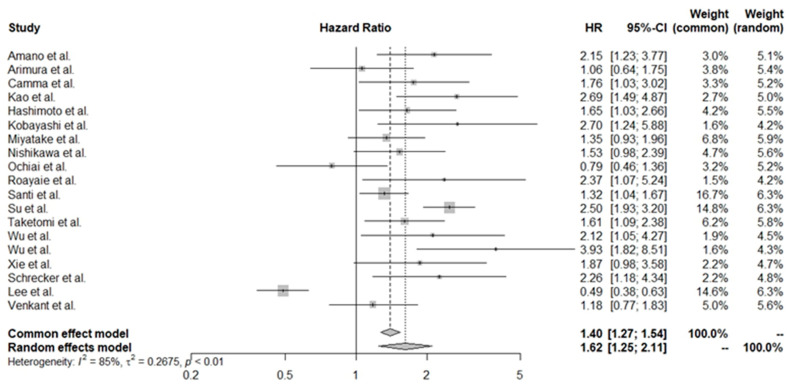
Pooled HR values for patients treated with curative intent [30,31,33,34,36,38,39,40,42,44,45,46,47,48,49,51,52,53,54].

**Figure 5 cancers-16-01293-f005:**
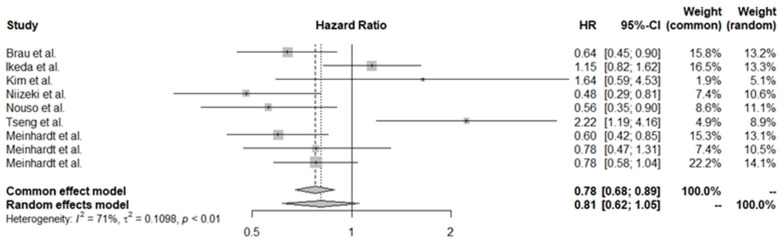
Pooled HR values for patients treated with non-curative intent [28,32,35,37,41,43,50].

**Figure 6 cancers-16-01293-f006:**
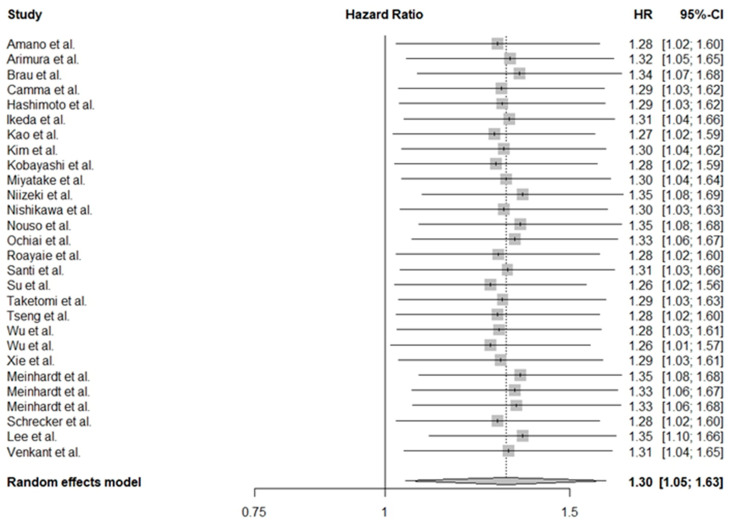
Influence analysis of included studies [28,30,31,32,33,34,35,36,37,38,39,40,41,42,43,44,45,46,47,48,49,50,51,52,53,54].

**Figure 7 cancers-16-01293-f007:**
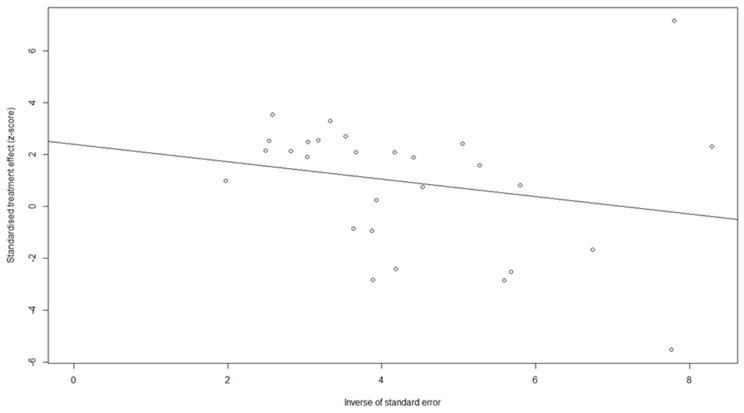
Egger funnel plot for the included studies.

**Table 1 cancers-16-01293-t001:** Baseline characteristics of the studies included in the meta-analysis.

Study	Reference	Country	Number of Patients (M/F)	Treatment	Child–Pugh (A/B/C)	PLT Cut-Off Value (10^9^/L)	Follow-Up (Years)	NOS
Amano et al.	[30]	Japan	127/24	Surgery	129/22	100	4.1	6
Arimura et al.	[31]	Japan	95/45	PEIT/TACE	77/48/15	80	NA	4
Bräu et al.	[32]	America	287/2	Surgery /RFA	113/134/42	100	NA	6
Cammá et al.	[33]	Italy	131/71	RFA	165/37	100	1.25	8
Hashimoto et al.	[34]	Japan	120/29	Surgery	NA	120	3.5	9
Ikeda et al.	[35]	Japan	122/46	TACE	86/82	75	2.8	5
Kao et al.	[36]	Taiwan	162/69	RFA	226/32	100	2.375	6
Kim et al.	[37]	Korea	39/13	TACE	16/23/13	150	0.4	6
Kobayashi et al.	[38]	Japan	146/53	Surgery	199	100	3.25	7
Lee et al.	[39]	Taiwan	1245/415	Surgery /RFA/TACE	830/349/149	118	5	7
Miyatake et al.	[40]	Japan	260/135	Surgery	317/74/4	100	3.5	7
Meinhardt et al.	[28]	America	221	Systemic treatment	A major	150	NA	4
			95					
			318					
Niizeki et al.	[41]	Japan	56/15	TACE	43/28	120	NA	5
Nishikawa et al.	[42]	Japan	217/151	RFA	70/162/100	100	3	8
Nouso et al.	[43]	Japan	116/41	RFA/TACE	157(C)	80	NA	7
Ochiai et al.	[44]	Japan	208/76	Surgery	273/11	110	3	6
Roayaie et al.	[45]	America	95/37	Surgery	132	150	3.2	7
Santi et al.	[46]	Italy	457/192	Surgery/RFA/TACE	477/172	100	3.3	9
Schrecker et al.	[47]	Germany	96/32	Surgery	126/2	100	4.6	8
Su et al.	[48]	Taiwan	152/36	Surgery	A major	100	5.8	8
Taketomi et al.	[49]	Japan	167/43	Surgery	158/52	150	2.4	7
Tseng et al.	[50]	Taiwan	48/34	Surgery /RFA/TACE	NA	100	NA	6
Wu et al.	[51]	Taiwan	104/57	RFA	A major	100	3.2	4
Wu et al.	[52]	China	79/7	Surgery	A major	100	NA	7
Venkant et al.	[53]	America	1411/686	Surgery	NA	150	NA	7
Xie et al.	[54]	China	408/79	RFA/TACE	NA	97	NA	6

Abbreviations: PEIT—Percutaneous ethanol injection therapy; RFA—Radiofrequency ablation; TACE—Transcatheter arterial chemoembolization; NOS—Newcastle–Ottawa scale.

**Table 2 cancers-16-01293-t002:** Characteristics of patient groups in included studies.

Study	Age (Years)	Race	Follow-Up	Disease Stage (BCLC)	Viral Infection	Alcohol Intake
Meinhardt et al. [28]	NA	NA	NA	NA	NA	NA
Amano et al. [30]	>18	NA	4.1 ± 3.1 years	NA	HCV	NA
Arimura et al. [31]	63.3 ± 8.54	NA	NA	NA	NA	NA
Bräu et al. [32]	52.2 (±8.0)–63.9 (±11.0)	White, Latino, Black, Asian	NA	A-D	HIV, HCV, HBV	+
Cammà et al. [33]	66.8 (±8.2) 67.4 (±6.9)	NA	15 months	A-B	HCV, HBV	+
Hashimoto et al. [34]	61.7	N/A	42.1 months	N/A	HCV, HBV	N/A
Ikeda et al. [35]	63 (45–80)	N/A	2.8 years	N/A	HCV, HBV	+
Kao et al. [36]	>18	N/A	28.5 ± 18.7 months	N/A	HCV, HBV	N/A
Kim et al. [37]	57 (35–80)	N/A	5 months	N/A	HCV, HBV	+
Kobayashi et al. [38]	62 (29–80)–67 (38–87)	N/A	3.3 years	N/A	HCV, HBV	+
Lee et al. [39]	N/A	N/A	5 years	A	HCV, HBV	N/A
Miyatake et al. [40]	58	N/A	1.3	N/A	HCV	N/A
Niizeki et al. [41]	65	N/A	N/A	N/A	HCV, HBV	N/A
Nishikawa et al. [42]	69.9 ± 9.0	N/A	N/A	N/A	HCV, HBV	N/A
Nouso et al. [43]	63	N/A	N/A	N/A	HCV, HBV	N/A
Ochiai et al. [44]	63.9	N/A	36 months	N/A	HCV, HBV	+
Roayaie et al. [45]	63.1 ± 10.5	N/A	37.5 months	0	HCV, HBV	+
Santi et al. [46]	67	N/A	38.6 ± 32.8 months	N/A	HCV, HBV	+
Schrecker et al. [47]	65 (34–81)	N/A	55.1 months	0-B	N/A	N/A
Su et al. [48]	61.5 (52.0–70.75)	N/A	69.8 months	N/A	HCV, HBV	N/A
Taketomi et al. [49]	60.7 ± 7.9	N/A	26.6 ± 22.0 months	N/A	N/A	N/A
Tseng et al. [50]	65.8 ± 9.6	N/A	4 years	0-B	HCV, HBV	N/A
Wu et al. [51]	67.5 ± 11.4	N/A	38.1 ± 20.8 months	N/A	HCV, HBV	N/A
Wu et al. [52]	>18	N/A	7 years	I–IV *	HBV	N/A
Venkat et al. [53]	64	Caucasian, African–American	N/A	N/A	N/A	N/A
Xie et al. [54]	52 ± 7.3–69 ± 3.7	N/A	N/A	N/A	HCV, HBV	N/A

*—Study accessed stage based on TNM classification. +—Included in the analysis.

**Table 3 cancers-16-01293-t003:** Subgroup Analysis (Random-Effects Model).

Covariates	Subgroup	No	HR	Heterogeneity		
				Ps	I^2^	Pa
Curative	No	9	0.7754 [0.6760; 0.8894]	0.115	70.5%	<0.001
	Yes	19	1.4006 [1.2717; 1.5426]	<0.001	84.7%	
Child–Pugh	≤50%	4	0.9099 [0.5312; 1.5587]	0.731	78.5%	0.349
	>50%	23	1.3921 [1.0847; 1.7865]	0.009	87%	
	No Data	1	1.1800 [0.7656; 1.8187]	0.453	0%	
HCV	≤50%	10	1.2986 [0.8032; 2.0995]	0.286	92.4%	0.883
	>50%	14	1.2797 [0.9778; 1.6748]	0.019	79.7%	
	No Data	4	1.4110 [1.0576; 1.8825]	0.072	32%	
PLT	≤100	17	1.6004 [1.2515; 2.0465]	<0.001	80.1%	0.028
	101–149	5	0.8207 [0.4778; 1.4099]	0.474	84.0%	
	150	6	1.0251 [0.7137; 1.4724]	0.893	77.4%	
Number of patients	≤200	15	1.5324 [1.1127; 2.1105]	0.009	82.2%	0.127
	>200	13	1.1016 [0.8341; 1.4548]	0.495	85.8%	

Ps—*p*-value for subgroups, Pa—*p*-value for all included.

## Data Availability

The data can be shared up on request.

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
