# Peer review of "Impact of Thrombocytopenia on Survival in Patients with Hepatocellular Carcinoma: Updated Meta-Analysis and Systematic Review"

_cancers, 2024, doi:10.3390/cancers16071293_

Round 1

Reviewer 1 Report

Comments and Suggestions for Authors

good review

1. line 127 - spelling of imaging is incorrect

2. Line 170- can we separate patients who were not able to get curative treatment due to low platelets and resultant low overall survival as a separate group than patients with low platelets who still were able to get curative treatment?

Zhang Z, Zhang Y, Wang W, Hua Y, Liu L, Shen S, Peng B. Thrombocytopenia and the outcomes of hepatectomy for hepatocellular carcinoma: a meta-analysis. J Surg Res. 2017 Apr;210:99-107. 

3. line201-203 - Including TACE in palliative treatment is older classification. Presently, segmental TACE and Y90 are done with curative intent.

4. Line 269, authors mention - well established association of platelet count with HCC outcome - should it be association instead of well established.

Author Response

We found the advice very helpful and have revised our manuscript accordingly. In the summary below, the numbering of our responses corresponds to the sections of particular comments. We have addressed each comment in a point-by-point manner.

Reviewer 1 :

good review

Response: We would like to thank the reviewer for his opinions and additional comments, which allowed us to improve our work.

  1. line 127 - spelling of imaging is incorrect

Response: We have revised our manuscript accordingly.

  1. Line 170- can we separate patients who were not able to get curative treatment due to low platelets and resultant low overall survival as a separate group than patients with low platelets who still were able to get curative treatment?

Zhang Z, Zhang Y, Wang W, Hua Y, Liu L, Shen S, Peng B. Thrombocytopenia and the outcomes of hepatectomy for hepatocellular carcinoma: a meta-analysis. J Surg Res. 2017 Apr;210:99-107. 

Response: We reviewed the data from the trials included in the analysis. Despite our best intentions, the studies data do not allow us to distinguish between these two groups of patients. We do not have access to the medical records of the included patients who were not treated because their platelet levels were too low. At the same time, based on our own clinical experience with this group of patients, only very low platelet counts disqualify the patient from radical treatment. However, this is a very interesting clinical question that leaves room for such analysis.

  1. line201-203 - Including TACE in palliative treatment is older classification. Presently, segmental TACE and Y90 are done with curative intent.

Response: We have added a sentence clarifying this issue in section 3.4. Pooled HR Values for Various Treatment Groups : "However, it is important to note that segmental TACE can currently be used with curative intent [55]. "

  1. Line 269, authors mention - well established association of platelet count with HCC outcome - should it be association instead of well established.

Response: We have revised our manuscript accordingly.

Reviewer 2 Report

Comments and Suggestions for Authors

This is a well-organized manuscript to perform meta-analysis on multiple HCC cohorts. The major concern will be the novelty of this manuscript. By integrating multiple cohorts, what are the additional discoveries compared with single cohort analysis? Here are some major suggestions.

1.        A summary table for the selected HCC cases (per study or merged studies) should be provided to show the distribution of age, gender, race, follow-up time, disease stage, and many risk factors, such as alcohol taken, cigarette, virus infection, etc.

2.        For the inclusion/exclusion criteria, will age, follow-up time, disease stage or risk factors to be taken into consideration?

3.        How to interpret the inconsistency/ diversity of different cohorts? For example, Figure 2 to 5 shows diverse Hazard Ratios among multiple cohorts? How to understand this results and draw an overall meta-conclusion?

4.        What are the major contributions/ innovations of this project? Compared with individual study, what additional conclusions can be drawn from the meta-analysis? Besides showing the hazard ratio per cohort at sub-population of HCC patients, what are the novel discoveries by integrating all the cohorts?

Comments on the Quality of English Language

Good quality

Author Response

We found the advice very helpful and have revised our manuscript accordingly. In the summary below, the numbering of our responses corresponds to the sections of particular comments. We have addressed each comment in a point-by-point manner.

Reviewer 2:

This is a well-organized manuscript to perform meta-analysis on multiple HCC cohorts. The major concern will be the novelty of this manuscript. By integrating multiple cohorts, what are the additional discoveries compared with single cohort analysis? Here are some major suggestions.

Response: We would like to thank the reviewer for his opinions and additional comments, which allowed us to improve our work.

  1. A summary table for the selected HCC cases (per study or merged studies) should be provided to show the distribution of age, gender, race, follow-up time, disease stage, and many risk factors, such as alcohol taken, cigarette, virus infection, etc.

Response: The following table can be included in the manuscript as a review comment, but the studies lack important data and we leave the final decision on inclusion to the reviewer.

Table 2. Characteristics of patient groups in included studies.

Study

Age (years)

Race

Follow-up

Disease stage (BCLC)

Viral infection

Alcohol intake

Meinhardt et al. [1]

N/A

N/A

N/A

N/A

N/A

N/A

Amano et al. [2]

>18

N/A

4.1 ± 3.1 years

N/A

HCV

N/A

Arimura et al. [3]

63.3 ± 8.54

N/A

N/A

N/A

N/A

N/A

Bräu et al. [4]

52.2 (±8.0)-63.9 (±11.0)

White, Latino, Black, Asian

N/A

A-D

HIV, HCV, HBV

+

Cammà et al. [5]

66.8 (±8.2) 67.4 (±6.9)

N/A

15 months

A-B

HCV, HBV

+

Hashimoto et al. [6]

61.7

N/A

42.1 months

N/A

HCV, HBV

N/A

Ikeda et al. [7]

63 (45-80)

N/A

2.8 years

N/A

HCV, HBV

+

Kao et al. [8]

>18

N/A

28.5 ± 18.7 months

N/A

HCV, HBV

N/A

Kim et al. [9]

57 (35–80)

N/A

5 months

N/A

HCV, HBV

+

Kobayashi et al. [10]

62 (29-80)-67 (38-87)

N/A

3.3 years

N/A

HCV, HBV

+

Lee et al. [11]

N/A

N/A

5 years

A

HCV, HBV

N/A

Miyatake et al. [12]

58

N/A

1.3

N/A

HCV

N/A

Niizeki et al. [13]

65

N/A

N/A

N/A

HCV, HBV

N/A

Nishikawa et al. [14]

69.9 ± 9.0

N/A

N/A

N/A

HCV, HBV

N/A

Nouso et al. [15]

63

N/A

N/A

N/A

HCV,HBV

N/A

Ochiai et al. [16]

63.9

N/A

36 months

N/A

HCV, HBV

+

Roayaie et al. [17]

63.1 ± 10.5

N/A

37.5 months

0

HCV, HBV

+

Santi et al. [18]

67

N/A

38.6 ± 32.8 months

N/A

HCV, HBV

+

Schrecker et al. [19]

65 (34–81)

N/A

55.1 months

0-B

N/A

N/A

Su et al. [20]

61.5 (52.0–70.75)

N/A

69.8 months

N/A

HCV, HBV

N/A

Taketomi et al. [21]

60.7 ± 7.9

N/A

26.6 ± 22.0 months

N/A

N/A

N/A

Tseng et al. [22]

65.8 ± 9.6

N/A

4 years

0-B

HCV, HBV

N/A

Wu et al. [23]

67.5 ± 11.4

N/A

38.1±20.8 months

N/A

HCV, HBV

N/A

Wu et al. [24]

>18

N/A

7 years

I-IV*

HBV

N/A

Venkat et al. [25]

64

Caucasian, African–American

N/A

N/A

N/A

N/A

Xie et al. [26]

52 ± 7.3-69 ± 3.7

N/A

N/A

N/A

HCV, HBV

N/A

*- Study accessed stage based on TNM classification.

  1. For the inclusion/exclusion criteria, will age, follow-up time, disease stage or risk factors to be taken into consideration?

Response: Of course, these are very interesting factors that can be taken into account in the analysis, but currently the data from the literature do not allow for such a precise breakdown. Due to the small number of PLT analyses in HCC therapy, the groups of patients included are heterogeneous and do not allow separate data for individual subgroups.

  1. How to interpret the inconsistency/ diversity of different cohorts? For example, Figure 2 to 5 shows diverse Hazard Ratios among multiple cohorts? How to understand this results and draw an overall meta-conclusion?

Response: Thank you for the interesting question. As the reviewer rightly points out, the HR is different in each of the subgroups we present, this is due to the analysis indicating different prognoses in different cohorts. Therefore, the HR estimate for the general population is HR >1 (HR = 1.30, 95% CI: 1.05-1.63), indicating a worse prognosis. Interestingly, the analysis of smaller cohorts, which we have subdivided according to treatment intent, already shows significant differences - patients treated palliatively with HR <1 (HR 0.81, 95% CI 0.62-1.05) have a completely opposite prognosis with low PLT levels and can therefore achieve a better OS with therapy.

  1. What are the major contributions/ innovations of this project? Compared with individual study, what additional conclusions can be drawn from the meta-analysis? Besides showing the hazard ratio per cohort at sub-population of HCC patients, what are the novel discoveries by integrating all the cohorts?

Response: Individual analyses of the impact of PLT on a patient's prognosis provide valuable information about the prognosis in a specific group of patients, for example, an analysis that considers only surgically treated patients. We have not seen a direct comparison of palliatively and radically treated patients in a trial. Because there are strict guidelines for the inclusion of patients in different treatments, we decided to check whether the thresholds for inclusion were really appropriate. For example, taking into account our results, it may be possible in the near future to include patients with much lower PLT levels, who currently cannot benefit from treatment, both in clinical trials and in everyday practice. The added value of our work is mainly the clinical implication, which may provide physicians with additional options for patients who are burdened and suffer from permanently reduced platelet parameters. There were also some inconsistencies in the results of the individual analyses. For example, in some analyses patients with reduced platelets had a better prognosis, and in others a worse prognosis. Our work points to a possible source of this discrepancy, namely the type of treatment.

Round 2

Reviewer 2 Report

Comments and Suggestions for Authors

Thanks for the great revision works that have been done by the authors. 

1.        Table 2 is a great summary. I would recommend to keep this table in the manuscript, or combine with Table 1, or at least keep it as a supplementary table.

2.        Please put your response to the Discussion section as a limitation or future work.

3.        Can you discuss this in the Result/ Discussion section, please?

4.        The novelty of the meta analysis should be one of the main point of this type of meta-analysis manuscript. I would suggest the authors to add and emphasize the novel discoveries in the Result/ Discussion section. The response looks great, but not reflected in the manuscript revision.

Author Response

Thank You for your previous important comments, as suggested we have added our previous response to the last paragraph regarding the study's limitations and possible directions for developing the topic. All suggestions and tables have been added to the latest version of the attached manuscript.

Kind Regards

Authors
